

# Zoonotic spillover and viral mutations from low and middle-income countries: improving prevention strategies and bridging policy gaps

Zulfqarul Haq[1,*], Junaid Nazir[2,3,*], Tasaduq Manzoor[3], Afnan Saleem[3], H. Hamadani[1], Azmat Alam Khan[1], Sahar Saleem Bhat[3], Priyanka Jha[2] and Syed Mudasir Ahmad[3]

[1] ICMR project, Division of Livestock Production and Management, Sher-e-Kashmir University of Agricultural Sciences and Technology of Kashmir, India, Srinagar, Jammu and Kashmir, India
[2] Department of Clinical Biochemistry, Lovely Professional University, Phagwara, Punjab, India
[3] Division of Animal Biotechnology, Faculty of veterinary Sciences, Sher-e-Kashmir University of Agricultural Sciences and Technology of Kashmir, India, Srinagar, Jammu and Kashmir, India
* These authors contributed equally to this work.

Corresponding author
Syed Mudasir Ahmad,
mudasirbio@gmail.com

## ABSTRACT

The increasing frequency of zoonotic spillover events and viral mutations in low and middle-income countries presents a critical global health challenge. Contributing factors encompass cultural practices like bushmeat consumption, wildlife trade for traditional medicine, habitat disruption, and the encroachment of impoverished settlements onto natural habitats. The existing "vaccine gap" in many developing countries exacerbates the situation by allowing unchecked viral replication and the emergence of novel mutant viruses. Despite global health policies addressing the root causes of zoonotic disease emergence, there is a significant absence of concrete prevention-oriented initiatives, posing a potential risk to vulnerable populations. This article is targeted at policymakers, public health professionals, researchers, and global health stakeholders, particularly those engaged in zoonotic disease prevention and control in low and middle-income countries. The article underscores the importance of assessing potential zoonotic diseases at the animal-human interface and comprehending historical factors contributing to spillover events. To bridge policy gaps, comprehensive strategies are proposed that include education, collaborations, specialized task forces, environmental sampling, and the establishment of integrated diagnostic laboratories. These strategies advocate simplicity and unity, breaking down barriers, and placing humanity at the forefront of addressing global health challenges. Such a strategic and mental shift is crucial for constructing a more resilient and equitable world in the face of emerging zoonotic threats.

## INTRODUCTION

Zoonotic spillover, described as the pathogen transmission from vertebrate animals to human beings, is an intricate biological process and remains a global public health concern (*Cross et al., 2019*). Many zoonotic viruses, including rabies, avian influenza, severe acute respiratory syndrome (SARS-1 and 2), Middle East respiratory syndrome coronavirus, West Nile virus, Ebola virus, Hendra virus, Hantaviruses, and Nipah virus, have emerged in human populations through such phenomena. The risk of these zoonotic infections is currently elevated, with three out of four new diseases being identified as zoonotic, as reported by the CDC (*Breedlove, 2022*). According to estimates, 60–75% of infectious diseases that emerge in humans are generally caused by zoonotic agents. The ongoing transmission of pathogens, such as Human Immunodeficiency Virus (HIV), Influenza-A virus (H1N1), Ebola virus, Rabies virus, respiratory syndrome coronavirus (CoV), and West Nile virus, are examples of these outbreaks (*Jones et al., 2008*). A zoonotic spillover may occur from a natural reservoir, directly or indirectly, from the virus in the environment or through an intermediate host. Pathogens are usually spread through contact with meat, blood, biofluids, contaminated surfaces, and aerosols (*Aguirre et al., 2020*). Zoonotic spillovers leading to disease outbreaks and pandemics drastically affect economic, social, political, and environmental systems (*Sánchez, Venkatachalam-Vaz & Drake, 2021*).

Spillover risk is also determined by the environment's density and distribution of reservoir hosts. Pathogenic transmission *via* aerosols and contaminated surfaces is more common in environments where different animal species are confined together (*Plowright et al., 2014*). It is also essential to consider how pathogen characteristics affect spillover risk since they determine how viable the pathogen is in the environment and how easily it can be transmitted between different hosts or vectors (*Wassenaar & Zou, 2020*).

Human activities like deforestation and the construction of settlements in sylvan environments have increased human interaction with different animal species, resulting in spillover events (*Morand, Owers & Bordes, 2014*; *Van Langevelde et al., 2020*) (Fig. 1). The exponential human population growth worldwide, resulting in the continuous loss of global biodiversity due to various human activities, hints towards more frequent spillover events in the coming years (*Ellwanger et al., 2020*; *Ramírez et al., 2020*). According to the recent World Health Organisation (WHO) report, it is theorized that the novel coronavirus likely originated in wildlife and spread to humans *via* bushmeat sold in Wuhan wet markets (*Maxmen, 2022*). Thus, it becomes clear that bush meat, wet markets, and zoonotic diseases all share a well-documented relationship. In many Asian, African, and South American (Amazonas) countries, it is an essential source of meat for poor people or has become a delicacy for urban people. Therefore, such countries may develop into a potential breeding ground for mutant viruses that could cause global pandemics in the future (Fig. 2).

Low and middle-income countries (LMICs), with a lower Gross National Income (GNI) *per capita*, grapple with economic challenges, leading to restricted access to resources and lower living standards. GNI classifications aid in recognizing global economic disparities,
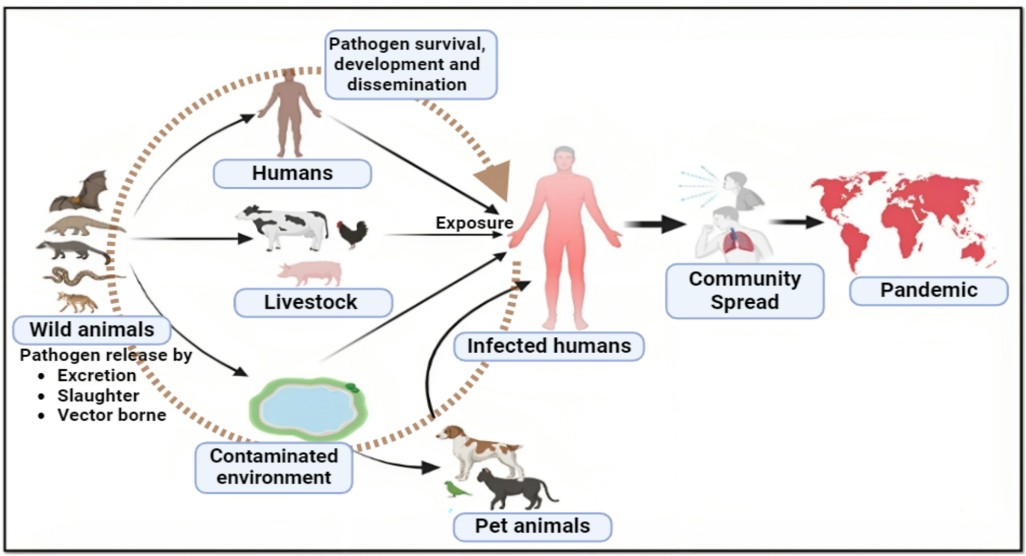

**Figure 1 Major pathways to zoonotic spillover.**

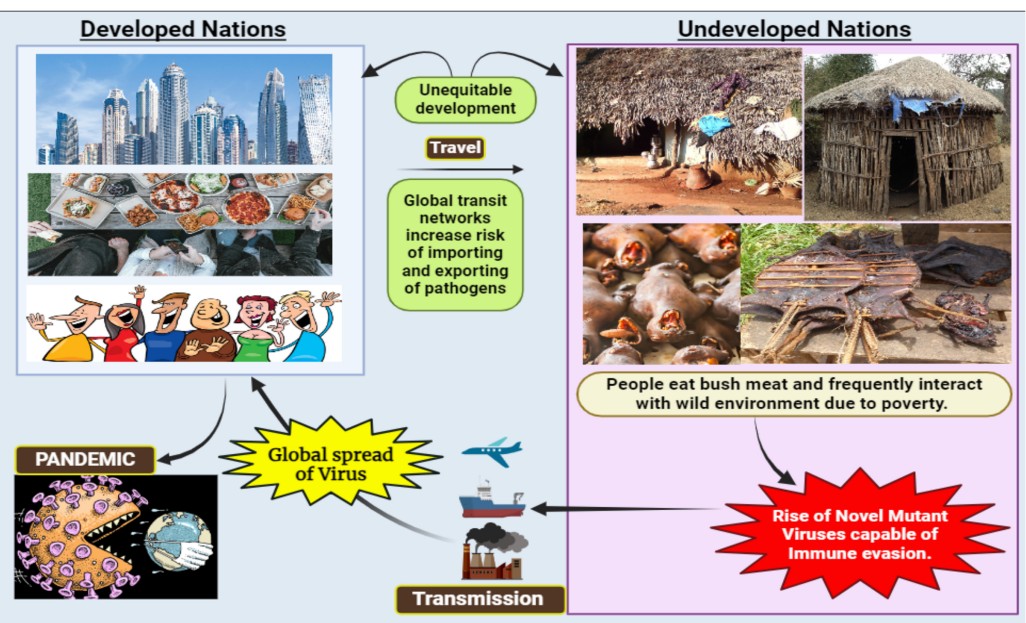

**Figure 2 Consequences of viral spillover from undeveloped countries to developed nations.**

shaping development strategies, and allocating resources based on a country's economic status. The vaccine gap signifies the inequality in vaccine access worldwide, driven by economic, logistical, and nationalistic factors (*Ahmed Ali et al., 2022*). This gap in low- and middle-income countries enhances the risk of unchecked virus propagation and mutation due to the absence of a proofreading mechanism in viral replication (*Ahmed Ali et al., 2022*). Unlike eukaryotic cells, these viruses lack a proofreading mechanism to prevent

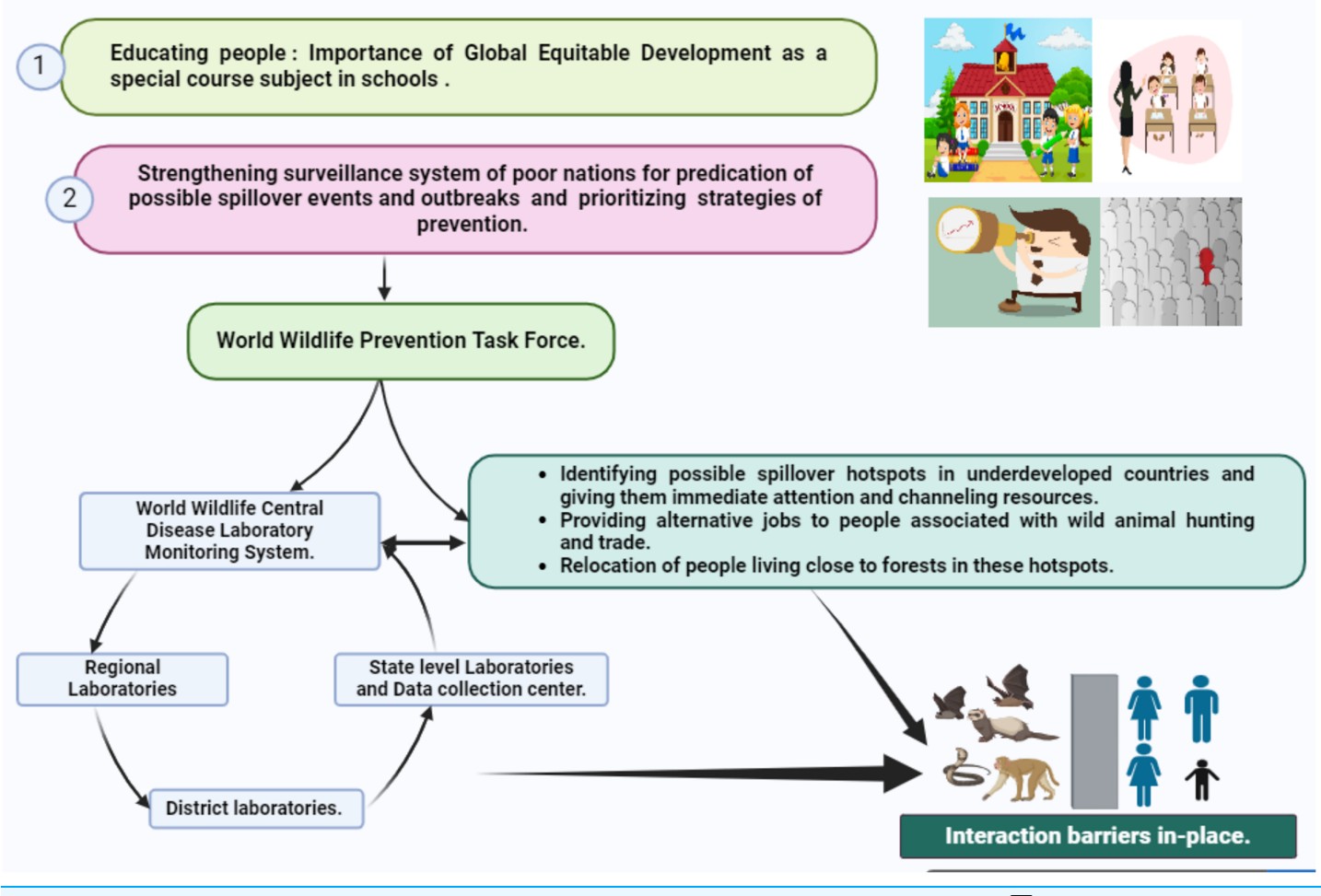

**Figure 3 Mitigation strategies for spillover events.**

replication errors, resulting in the production of mutant viruses—a potential global time bomb (*Aparicio-Domingo et al., 2021*).

Spillover infectious pathogens must overcome multiple barriers, including those related to diseases, humans, animals, and the environment before they can spread to susceptible humans (*Plowright et al., 2017*; *Worsley-Tonks et al., 2022*). Many of these barriers are prevalent in underdeveloped and developing nations in Asia, Africa, and the South American countries rendering them particularly vulnerable to spillover events (*Worsley-Tonks et al., 2022*). Several highly deforested areas in the Amazon can already be considered at high risk for infectious disease spillovers, and if tropical deforestation continues to accelerate, these high-risk areas could expand (*Alagona et al., 2020*). Even a small amount of deforestation, especially if it occurs in pristine, previously undisturbed areas, could have a disproportionate impact on the chances that a zoonotic virus such as COVID-19, Ebola, or bird flu could emerge (*Giroh & Nachandiya, 2020*; *Reperant & Osterhaus, 2017*). Factors such as habitat degradation, the live wild animal trade, and the resultant nutritional and physiological stress on wildlife can lead to immunosuppression,

pathogen shedding, and increased susceptibility to new diseases (*Plowright et al., 2021*). Moreover, evolving dynamics in these nations, driven by factors like changing land use, land management, climate change, and human-animal interactions, are accelerating pathogen distribution and increasing the likelihood of new diseases emerging annually (*Baker et al., 2022*).

Therefore, assessing the potential for zoonotic diseases, especially in developing and undeveloped countries, is imperative. Such evaluations can offer insights for interventions aimed at reducing the risk of spillover events. This review explores the mechanisms of spillover and mutation, both generally and within the context of the developing world. It highlights challenges, deficiencies in current global policies, and practical mitigation strategies tailored to these regions (Fig. 3).

## SURVEY METHODOLOGY

We identified review articles and peer-reviewed published reports on four electronic databases: Google Scholar, PubMed, Web of Science, and Google Search, using the following search strings: zoonotic spillover, viral mutations, bushmeat hunting in Asian, African and South American countries, prevention strategies and global health policies. These databases were chosen for their coverage of a vast range of topics and scientific disciplines, and the timeline from 1995 to 2023 was chosen. However, only articles that provided full text in English were considered for analysis. The search yielded 327 articles. Twenty-seven articles that did not provide full text in English were omitted. Thereafter, we manually screened the resulting article abstracts to include only primary literature that explicitly established zoonotic pathogens (hereafter risk assessment studies) associated with hunting or consumption of bush meat in poor Asian, African, and South American countries, spillover, mutation, surveillance, vaccination and vaccination gap, current policies, their deficiencies, and corrective measures. Articles were excluded if they met any of the following criteria: (1) not a review article (single case study report not included) (2) an article focused solely on financial spillover aspects that were not included; (3) an article with a narrow scope *i.e.*, spillover limited to a single pathogen or host; or (4) articles based on spillover from companion animals (*e.g.*, pet dogs, cats) to humans.

All downloaded articles were read in full detail, and 85 articles were finally selected, on which an outline was formed after discussion with the authors. From these articles text segments for subtopics were identified, along with the gaps in policies. The researchers also snowballed additional data sources, obtaining relevant articles from reference lists of articles identified through our systematic search. Supplementary data were gathered from country reports, newsletters, commentaries, and policy briefs from the Food and Agriculture Organization (FAO) and the World Organisation for Animal Health (WOAH) because it was determined that some of the relevant sources were not published in peer-reviewed academic journals because they were policy articles rather than empirical studies. The extracted information was discussed narratively to explore the aim of the study, arranged chronologically, and compiled together. The gaps in policies

were written after multiple discussions with the authors until common conscience points were reached.

## Spillover and surveillance

### Spillover mechanisms and risks

As the potential for spillover events from domestic or wild animals to humans continues to grow, it is crucial to comprehend the mechanisms underlying the transmission of viruses from animals to humans. These mechanisms can vary depending on the specific viral pathogen and the host species involved. The persistence of viruses in reservoir hosts can occasionally lead to mutations or recombination events, giving rise to new viral variants (*Dennehy, 2017*; *Snedden et al., 2021*). These mutations, particularly in the receptor-binding domains (RBDs) of spike proteins in coronaviruses, can impact their specificity, interaction capabilities, and binding affinity, ultimately influencing their ability to cross species boundaries (*Zhou et al., 2020*).

For a virus to survive and propagate, it must navigate the host's immune defence or find a new susceptible host (often referred to as a second or final host). The presence of shared cellular receptors is a critical factor in determining a host's susceptibility to viral infections. When different animal species share these receptors, they become susceptible to the risk of viral spillover (*Kuchipudi et al., 2021*). The presence of shared receptors in reservoir hosts, intermediate hosts, and humans, along with other animals, increases the risk of spillover infections (*Ye et al., 2020*).

The receptor-binding domains of viruses play a pivotal role in facilitating their interaction with host receptors, binding to host cells, and causing disease. For example, the spike protein RBD of coronaviruses, including SARS-CoV-2, recognizes Angiotensin-Converting-Enzyme-2 receptors found in various animal and human tissues, rendering various species susceptible to viral spillover (*Dhama et al., 2020*). This spillover can lead to the emergence of diseases, outbreaks, or even pandemics (*Rodriguez-Morales et al., 2020*). When a viral outbreak occurs in a second host, it triggers an immune response, which can result in illness or death, or it may lead to the establishment of a balance between the host's immune system and the pathogen. The persistence of pathogens in animals, humans, or the environment poses a continual risk for spillover events and is a key factor driving the potential for future pandemics. Therefore, regular surveillance is essential to develop effective strategies for combating these risks and minimizing the threat they pose to public health (*Shah et al., 2020*).

### Significance of surveillance

Surveillance functions as an early warning system by collecting data on viruses found in animal populations, allowing researchers to identify potential threats and assess the risk of spread into human populations (*Yadav & Akhter, 2021*). This proactive approach is critical for preventing the emergence of new diseases and minimising the societal impact of those diseases. Furthermore, surveillance provides critical information for assessing risk factors associated with future epidemics and pandemics. Understanding the prevalence of viruses in animals and their ability to infect humans provides the foundation for risk assessment

(*Grange et al., 2021*). By identifying high-risk areas and vulnerable species, measures can be put in place to reduce the likelihood of future pandemics. While many developed nations have established robust surveillance systems, low and middle-income countries face numerous challenges in implementing effective viral disease surveillance.

### Viral disease outbreaks in low and middle-income countries: the impact of limited surveillance

Understanding the consequences of low surveillance capacity is crucial in recognizing the vulnerability of countries to viral outbreaks. The following timeline highlights instances where inadequate surveillance contributed to the escalation of viral diseases in low and middle-income countries.

*Nipah virus outbreak in India (2001, 2007, and 2018)*
The 2018 outbreak of the Nipah virus in Kerala, India, underscores the crucial necessity for a comprehensive global surveillance system targeting *henipaviruses* found in bats—the primary reservoir hosts for Nipah virus and other related pathogens (*Arunkumar et al., 2019*; *Singh et al., 2019*). The Nipah virus, belonging to the *henipavirus* genus within the paramyxovirus family, induces severe illness in humans, featuring stuttering chains of transmission, and is recognized as a potential global pandemic threat.

Commencing in May 2018, the Nipah virus outbreak in Kerala was notably distant, exceeding 1,800 km from the locations of previous outbreaks in eastern India in 2001 and 2007 (*Mathew et al., 2021*). Unfortunately, the absence of adequate surveillance measures, coupled with delayed reporting, impeded the timely identification of the virus during its early stages. This lapse in detection contributed to elevated mortality rates and an extended duration of the outbreak, emphasizing the critical importance of early identification and prompt implementation of containment measures (*Dramowski et al., 2017*). To address this surveillance gap, it is imperative to enhance global monitoring efforts to promptly identify and mitigate the spread of emerging infectious diseases, particularly those originating from wildlife reservoirs (*Cunningham, Daszak & Wood, 2017*; *Morse, 2012*).

*Avian influenza (H5N1) in Southeast Asia (2003–2005)*
Since the initial occurrence of the highly pathogenic avian influenza virus (HPAIV) H5N1 in August 2003, Southeast Asian countries, including Cambodia, Laos, Malaysia, Myanmar, Indonesia, Thailand, and Vietnam, have all grappled with recurring outbreaks of this infectious disease (*Edler, 2006*). This alarming trend has led to a surge in human infections, precipitating a profound public health crisis with significant economic repercussions.

Despite concerted efforts to control the spread of HPAIV through strategies such as weak surveillance, vaccination, and stamping out, certain Southeast Asian nations continue to grapple with persistent outbreaks (*Eagles et al., 2009*). The region has witnessed the circulation of various H5N1 strains since the initial outbreaks in 2003 (*Worsley-Tonks et al., 2022*). Although the precise origin of the initial outbreaks in domestic poultry remains unknown, the ongoing proliferation of the disease in the region

can be primarily attributed to the movement of domestic poultry and poultry products, as well as human activities.

Addressing the challenges posed by HPAIV in Southeast Asia necessitates a multifaceted approach that includes bolstering surveillance capabilities, implementing effective vaccination protocols, and devising strategies to curtail the movement of infected poultry and products (*Eagles et al., 2009*; *Edler, 2006*). Additionally, efforts to enhance public awareness and collaboration across borders are imperative to mitigate the impact of this persistent threat to both public health and the regional economy.

### Ebola outbreak in West Africa (2014–2016)

The Ebola virus disease (EVD) outbreak that occurred from 2014 to 2016 in the West African sub-region emerged as a major global epidemic, intensifying international concerns in the past decade (*Kamorudeen, Adedokun & Olarinmoye, 2020*). Within this timeframe, the 2014–2016 epidemics of ebolavirus were particularly severe, presenting alarming case fatality rates and causing significant socioeconomic impact in the affected countries (*Kamorudeen, Adedokun & Olarinmoye, 2020*). A crucial contributing factor to the severity of the outbreak was the limited healthcare infrastructure and low surveillance capabilities in the affected nations. Recognizing these deficiencies is essential for understanding the root causes of the outbreak.

In light of these challenges, there is an urgent imperative to develop and implement an active early warning alert and surveillance response system. This proactive approach is crucial for effectively addressing outbreaks and controlling the spread of emerging infectious diseases (*Kock et al., 2019*; *Tambo, Ugwu & Ngogang, 2014*). By enhancing surveillance capabilities and establishing early warning mechanisms, the international community can better respond to and mitigate the impact of future outbreaks, ultimately safeguarding public health on a global scale.

### Yellow fever outbreak in Angola (2015–2016)

An outbreak of yellow fever was identified in Angola in late December 2015, and its confirmation came from the National Institute for Communicable Diseases in South Africa and Institute Pasteur of Dakar, Senegal on January 20, 2016 (*Rudnicka et al., 2020*). Subsequently, a notable surge in the number of cases occurred. Between December 5, 2015, and September 22, 2016, a total of 4,143 suspected cases were reported. Among them, 884 cases were laboratory-confirmed, resulting in 121 deaths and a case fatality rate of 13.7% (*Rossetto & Luna, 2016*).

The analysis of the outbreak highlighted a critical issue: inadequate surveillance systems struggled to effectively monitor and contain the virus's spread. This deficiency led to elevated fatality rates and posed challenges in distributing vaccines due to delayed detection (*Rossetto & Luna, 2016*). The incident underscores the importance of robust surveillance systems for timely intervention and response in managing infectious diseases, particularly in regions vulnerable to such outbreaks.

*Chikungunya outbreak in India (2006)*

Chikungunya, caused by the Chikungunya virus, has recently emerged as a significant public health concern in the Indian Ocean Islands and India. In 2006 alone, an estimated 1.38 million people in southern and Central India developed symptomatic disease (*Kalantri, Joshi & Riley, 2006*). However, the actual incidence may have been higher but was likely underreported due to the lack of accurate reporting mechanisms.

This tropical disease is transmitted by the Aedes mosquito and is characterized by symptoms such as fever, headache, rashes, and debilitating arthralgia. Initially believed to be non-fatal and self-limiting, Chikungunya has taken on a more severe form with reported cases involving central nervous system involvement and fulminant hepatitis (*Bonn, 2006*).

The first outbreak of the Chikungunya virus in Asia was reported in Thailand in 1958, and it re-emerged in Java, Indonesia, between 2001 and 2003 after a gap of 20 years (*Lien et al., 2018*). Unfortunately, limited surveillance infrastructure contributed to the delayed identification of the outbreak, resulting in widespread transmission and posing challenges in implementing effective control measures (*Bonn, 2006*; *Kalantri, Joshi & Riley, 2006*; *Rossetto & Luna, 2016*; *Kalantri, Joshi & Riley, 2006*). This emphasizes the critical need for improved surveillance systems to enable early detection and rapid response to emerging infectious diseases, preventing their unchecked spread and mitigating their impact on public health.

*Measles outbreak in Madagascar (2018–2019)*

In 2019, Madagascar confronted its most extensive documented measles outbreak in history. Spanning from September 2018 to January 2020, the outbreak affected nearly 225,000 individuals, resulting in slightly over 1,000 fatalities (*Meckawy et al., 2022*). Madagascar, being one of the world's poorest nations with one of the least funded health systems, faced significant challenges in managing the outbreak due to its weak healthcare infrastructure and struggling surveillance systems (*Finnegan et al., 2020*).

*COVID-19 pandemic (2019-present)*

The global impact of COVID-19 surged like a tsunami. The first cases were identified in December 2019 in Wuhan, China, and by March 2020, infections had permeated worldwide (*De Nicola et al., 2022*). Throughout the COVID-19 pandemic, surveillance systems have predominantly focused on reporting deaths, neglecting to monitor the broader spectrum of health implications and long-term consequences associated with the virus. The reporting has been further constrained, with only positive tests being counted, omitting a comprehensive tracking of all cases. Within 3 months of the first reported cases, COVID-19 had reached nearly 90% of WHO member states, with only 24 countries, primarily LMICs, not reporting cases as of March 30, 2020 (*Li et al., 2021*; *Solís Arce et al., 2021*). The limited testing capacity in these countries suggests a potential lack of capability to detect and identify cases early through sentinel and case-based surveillance.

Underestimating the number of cases poses a challenge to the effective control of severe acute respiratory syndrome coronavirus 2 (SARS-CoV-2). There is an urgent need for a

universal agreement and implementation of surveillance case definitions for both infection and recovery (*Koh & Goh, 2020*; *Senia et al., 2021*). Initial underreporting and delayed recognition of human-to-human transmission contributed to the rapid global spread, overwhelming healthcare systems, and significant socio-economic impacts (*Alwan, 2020b*). A universal surveillance case definition for recovery from COVID-19 is still absent. Many individuals experience prolonged symptoms, ill health, and reduced functionality for months, even without hospitalization for SARS-CoV-2 infection (*Alwan, 2020b*; *Koh & Goh, 2020*; *Li et al., 2021*). It is imperative to transition long-haul COVID from anecdotal to routine quantification and monitoring, aligning with the current practices for reporting deaths and positive tests (*Alwan, 2020a*).

### Strengthening global health: WHO's international programs for viral disease surveillance in low and middle-income countries for Asia and Africa

In an interconnected world, the threat of infectious diseases knows no borders. Recognizing the importance of global collaboration in disease surveillance and response, the WHO has been at the forefront of international efforts to strengthen healthcare systems in low and middle-income countries, particularly in Asia and Africa (*Heymann & Rodier, 2001*). Following are the key international programs initiated by the WHO to enhance viral disease surveillance in these regions, aiming to prevent, detect, and respond to outbreaks more effectively.

#### Global influenza surveillance and response system

The Global Influenza Surveillance and Response System (GISRS) is a pivotal component in the worldwide battle against influenza and other respiratory viruses (*Chadha et al., 2020*; *Hay & McCauley, 2018*; *Sanicas et al., 2014*). Operating through a network of National Influenza Centers and Collaborating Centers, this program plays a crucial role in promoting real-time data sharing, facilitating the exchange of samples, and fostering collaborative research. Particularly in economically disadvantaged and low and middle-income countriesacross Asia and Africa, GISRS actively contributes to the establishment and enhancement of surveillance systems (*Chadha et al., 2020*; *Sanicas et al., 2014*). By doing so, it enables the prompt detection of emerging influenza strains and supports the development of effective vaccines (*Broor et al., 2020*; *Ziegler, Mamahit & Cox, 2018*).

#### Global outbreak alert and response network

The Global Outbreak Alert and Response Network (GOARN) represents a worldwide alliance of institutions and networks dedicated to the swift identification, confirmation, and response to infectious disease outbreaks (*Imamura & Oshitani, 2021*). In its role through GOARN, the WHO extends support to countries in Asia and Africa. This support includes providing technical expertise, deploying response teams, and coordinating resources during public health emergencies (*Heymann, 2017*). By adopting this collaborative approach, GOARN ensures a rapid and effective response to contain the spread of viral diseases, particularly in resource-limited settings (*Ansell, Sondorp & Stevens, 2012*; *Mackenzie et al., 2014*).

*Global health security agenda*

The Global Health Security Agenda (GHSA) stands as a collective initiative aimed at fortifying global health security through the enhancement of capacities in countries to prevent, detect, and respond to infectious disease threats (*Organization, 2009*). In Asia and Africa, the WHO actively supports GHSA by aiding countries in the formulation and execution of national action plans, conducting risk assessments, and bolstering laboratory capacities. This multifaceted strategy is designed to cultivate resilient health systems capable of effectively managing public health emergencies, including those induced by viral diseases (*Paranjape & Franz, 2015*; *Wolicki et al., 2016*).

*Emergency medical teams*

Emergency medical teams (EMTs) are delineated by the WHO as "groups of health professionals and supporting staff outside their country of origin, aiming to provide health care specifically to disaster-affected populations" (*Katz et al., 2014*). The WHO's EMT initiative stands as a pivotal player in responding to health emergencies, particularly in impoverished and low and middle-income countries. Comprising specialized healthcare professionals and support staff, EMTs are swiftly deployed to areas affected by outbreaks. In Asia and Africa, the WHO collaborates with countries to establish and train EMTs, ensuring a prompt and well-coordinated response to viral disease outbreaks (*Foreign Medical Team Working Group, 2013*; *Hamilton, Södergård & Liverani, 2022*). The WHO international programs dedicated to viral disease surveillance in economically disadvantaged and low and middle-income countries across Asia and Africa exemplify a steadfast commitment to achieving global health equity (*Arziman, 2015*; *Goutard et al., 2015*). These initiatives, characterized by fostering collaboration, delivering essential technical assistance, and mobilizing critical resources, are designed with the overarching goal of fortifying healthcare systems. Moreover, they seek to elevate surveillance capabilities, ultimately aiming to mitigate the impact of infectious diseases on vulnerable populations ('Emerging and Re-Emerging Infectious Diseases', *Heymann & Rodier, 2001*). As the world confronts persistent health challenges, the imperative for sustained support for these programs becomes even more apparent. This ongoing backing is vital for not only reinforcing the global health landscape but also for creating a safer and more resilient environment in the face of emerging and recurrent health threats (*Ma et al., 2022*; *Tong et al., 2022*). The WHO's proactive approach to international collaboration in disease surveillance signifies a collective commitment to achieving health equity on a global scale (*Rogers Van Katwyk et al., 2023*; *World Health Organization (WHO), 2021*).

### Utilizing animals as sentinels: a global approach to early warning and surveillance for infectious diseases

In 2006, the WHO, FAO, and WOAH jointly established a global early warning and response system dedicated to monitoring and reporting disease outbreaks in public health and animal health systems (*World Health Organization, 2006*; *Zhao et al., 2021*). This collaborative surveillance system leverages the strengths of these global agencies to improve public and animal health by mitigating the incidence and impact of emerging

infectious diseases in both animals and humans. Contributors worldwide actively participate by providing real-time outbreak information, conducting risk assessments, and supporting the forecasting, prevention, and control of emerging diseases (*Halliday et al., 2017*). Given the coexistence of animals and humans in shared environments, where animals spend more time outdoors, they become valuable indicators for surveillance due to increased vulnerability to exposure risks (*Oeschger et al., 2021*; *Rabinowitz et al., 2005*). The comparatively shorter lifespan of animals enables the early detection of diseases following exposure to infectious agents, making them effective monitors for known and novel pathogens with outbreak potential. Animals have been implicated in over 60% of emerging infectious diseases in humans (*Hendricks, Mark-Carew & Conley, 2017*). Observing signs of illness, mortality, biomarkers, or sentinel events in animals offers a valuable approach for detecting, monitoring, or predicting environmental health hazards, human health hazards, or bioterrorism threats (*Shan Neo & Tan, 2017*).

Deaths in crows caused by the West Nile Virus, the clinical manifestation of swine flu in pigs due to H1N1, avian flu in ducks and other aquatic birds due to H5N1, and the identification of coronaviruses in bats all reinforce the proposition of employing animals as sentinels for surveillance (*Halliday et al., 2007*). Given their bio-accumulative nature, mussels, clams, and oysters can function as surveillance tools for both infectious and toxic chemicals. For instance, *Anomalocarids brasiliana* can be used to predict coliform contamination in shellfish harvesting systems (*Lima-Filho et al., 2015*). Canaries, known for their sensitivity, act as sentinels for poisonous gases, particularly carbon monoxide. Livestock such as sheep and cows are more susceptible than humans to *Bacillus anthracis*, and rodents and dogs are affected by *Yersinia pestis* (*Rabinowitz et al., 2006*). As both these infectious agents are categorized as potential bioterrorism agents, animals can provide an early warning about these harmful threats. Animals, therefore, serve as valuable surveillance tools for veterinarians, ecosystem health professionals, and researchers in monitoring and predicting various health hazards (*Shan Neo & Tan, 2017*).

Beyond observing disease manifestation and the bioaccumulation of toxic chemicals or infectious pathogens, it is crucial to conduct serological, microbiological, and molecular investigations to evaluate antibodies against infectious agents or their genetic material in domestic or wild animals that can impact humans (*van Zyl, 2022*). This approach not only provides information about animal exposure or the presence of pathogens but also indicates that the pathogen is circulating in the environment and has the potential to infect humans upon contact, as noted by *Keesing & Ostfeld (2021)*. The harboring of infectious agents by domestic and wild animals elevates the risk of human infections and may contribute to environmental contamination by these contagious agents (*Plowright et al., 2021*). Therefore, comprehensive surveillance efforts are instrumental in predicting the risk of disease and its potential spread, which could lead to an epidemic or pandemic.

Critical policy gaps that impede implementation of international programs for viral disease surveillance in low and middle-income countries (*Grover et al., 2020*; *Hendricks, Mark-Carew & Conley, 2017*; *Plowright et al., 2021*; *Rabinowitz et al., 2005*; *Wang et al., 2021*).

*Limited resources*

Low and middle-income countries often struggle with limited financial and human resources, hindering their ability to establish and maintain comprehensive surveillance systems. Insufficient funding for equipment, training, and personnel can result in suboptimal surveillance infrastructure.

*Infrastructure gaps*

In many low and middle-income countries, the lack of robust healthcare infrastructure poses a significant obstacle to effective viral surveillance. Limited laboratory facilities, inadequate transportation networks, and a shortage of skilled healthcare workers impede timely and accurate disease detection and reporting.

*Inadequate training and capacity building*

The shortage of trained professionals in epidemiology, virology, and laboratory diagnostics contributes to gaps in surveillance. Building local capacity through training programs and knowledge transfer is essential for strengthening surveillance systems.

*Neglected tropical diseases*

Neglected tropical diseases (NTDs) are a group of infectious diseases that primarily affect populations in low and middle-income countries (like chagas, leishmaniasis, African trypanosomiasis, schistosomiasis, cysticercosis, echinococcosis, dengue, rabies, ascariasis, Ebola and Zika viruses as well as other NTDs) (*Bhutta et al., 2014*). These diseases often go unnoticed due to limited surveillance efforts, exacerbating their impact on public health. Prioritizing surveillance for NTDs is crucial for their early detection and control (*Lozano et al., 2012*).

## Importance of global vaccination

To end the COVID-19 pandemic, a global rollout of vaccines promoting herd immunity is the most effective way out (*Agarwal & Reed, 2021*). To stop new variants or mutations in SARS-CoV-2, we must stop the virus from replicating globally, or a new variant will shatter all our preparedness in terms of vaccines developed so far useless. It is imperative to recognize that no one is safe until everyone is safe, emphasizing the need for equitable vaccine distribution to all corners of the globe. Unvaccinated people act as a mixing and multiplication vessel for the virus, leading to new variants (*Angius, Pala & Manzin, 2021*).

The interconnected nature of our world, characterized by frequent international travel and trade, makes infectious diseases highly transmissible across borders (*Suk et al., 2014*). A robust global vaccination strategy not only safeguards against the current pandemic but also fortifies our defences against potential future health crises (*Angius, Pala & Manzin, 2021*). By investing in a comprehensive vaccination infrastructure and fostering international collaboration, we create a shield against emerging threats, and low income enhancing our collective ability to respond swiftly and effectively to new infectious diseases that may arise (*Ahmed Ali et al., 2022*; *Reperant & Osterhaus, 2017*).

Furthermore, the economic and societal benefits of global vaccination cannot be overstated. The toll of a pandemic extends beyond the immediate health impact, affecting

economies, education systems, and social structures (*Goutard et al., 2015*; *Worsley-Tonks et al., 2022*). By achieving widespread immunity through global vaccination efforts, we not only save lives but also promote economic recovery and stability. Access to vaccines enables societies to reopen, facilitates the return to normalcy, and protects the livelihoods of individuals and communities. A resilient global vaccination infrastructure not only addresses the current health crisis but also contributes to building a more robust and adaptable framework for handling infectious diseases in the future, fostering a healthier and more secure world for generations to come (*Goutard et al., 2015*; *Halliday et al., 2017*).

### Global unity for equitable vaccination: urgent call to action for fair distribution and prevention of new SARS-CoV-2 variants

*International organizations' emphasis on fair distribution*
Various international organizations and leaders have stressed the importance of fair vaccine distribution. The WHO Director-General, Dr. Tedros Adhanom Ghebreyesus, has been vocal about the necessity of ensuring that vaccines reach every part of the world (*Keestra et al., 2022*). Recognizing that the virus knows no borders, the global community must unite in the fight against COVID-19, and this starts with fair access to vaccines.

*Encouraging high-income countries to share surplus vaccines*
The 'vaccine gap' faced by the undeveloped world in this COVID-19 pandemic is not new to human history. There have been uneven vaccine distributions in prior pandemics also (H1N1 recently) (*Agarwal & Reed, 2021*; *Alwan, 2020b*; *Giroh & Nachandiya, 2020*). Efforts should be redoubled to encourage governments of high-income countries to share surplus vaccines with LMICs. This approach can bridge the vaccine gap and ensure that vulnerable populations have access to life-saving vaccines (*Choudhary, Choudhary & Pervez, 2023*; *Turer et al., 2022*). A collaborative and generous sharing of resources can expedite the global vaccination process, bringing us closer to ending the pandemic on a global scale. A global allocation mechanism will be developed jointly by the Coalition for Epidemic Preparedness Innovations, the Global Alliance for Vaccines and Immunization, and WHO for the COVID-19 Vaccine Global Access. Facility to avoid a repeat of the H1N1 situation is currently the only hope for underdeveloped world (*Wouters et al., 2021*).

*Strengthening healthcare infrastructure in low and middle-income countries*
Beyond vaccine distribution, addressing healthcare disparities in low and middle-income countries is essential. Improving primary healthcare facilities and enhancing surveillance capabilities in these regions are critical for early detection and swift response to new outbreaks (*Goutard et al., 2015*; *Worsley-Tonks et al., 2022*) A holistic approach that combines vaccination efforts with strengthened healthcare systems is key to building lasting resilience against infectious diseases.

## Analyzing global legal frameworks in zoonotic disease prevention: strengthening international strategies for a sustainable future

### International legal frameworks

Recognizing the vital role of internationally recognized legal frameworks in mitigating zoonotic infectious diseases, this analysis delves into key instruments such as the International Health Regulations, the CITES Convention (Convention on International Trade in Endangered Species of Wild Flora and Fauna), and the Paris Agreement (*de Souza et al., 2020*; *Gupta et al., 2011*; *Hassell et al., 2017*). These frameworks have made substantial contributions to global efforts aimed at reducing the transmission of diseases from animals to humans. Additionally, we acknowledge the significance of entities like the WOAH and the Intergovernmental Science-Policy Platform on Biodiversity and Ecosystem Services (IPBES), proposing avenues for enhancement within these structures (*Keusch et al., 2022*; *Tajudeen et al., 2022*).

However, despite the strides made in zoonotic disease prevention, concerns persist among global health researchers regarding the escalating frequency of zoonotic pandemics, as highlighted by *Das Neves (2020)*. The analysis underscores the need for a more nuanced understanding of the evolving dynamics between human and animal ecosystems, particularly in developing and underdeveloped regions of Asia, Africa, and South America. Elevated levels of interaction and spillover events in these areas raise alarms about potential future pandemic risks, as noted by the IPBES (*Alimi et al., 2021*; *Marrana, 2022*; *ten Have, 2022*). We aim not only to acknowledge the existing contributions of legal frameworks but also to identify areas for improvement, fostering international cooperation and sustainable strategies to address the intricate challenges posed by zoonotic diseases on a global scale.

### Challenges in policy implementation

In undeveloped and developing countries, where sustainable hunting serves as a vital source of income and nutrition, poorly considered restrictions on wildlife hunting, consumption, and trade can disproportionately harm communities (*Tylianakis et al., 2021*). Although legal frameworks exist to tackle zoonotic diseases, challenges persist in implementing preventive measures. The International Health Regulations' focus on response measures falls short of addressing the underlying drivers of disease emergence (*Tylianakis et al., 2021*). In light of the ongoing threats posed by climate change and ecological destruction, there is a critical need to transform prevention approaches. It is imperative to shift the emphasis from solely detection and response to comprehensive prevention measures to mitigate the risk of future pandemics. Approximately 1/20[th] of the cost of human lives lost each year to emerging viral zoonoses can be attributed to primary prevention (*Bernstein et al., 2022*). Despite the acknowledgment of upstream drivers of zoonotic disease emergence in the G7 Carbis Bay Health Declaration (*Stern, 2021*), operationalizing and financing prevention-oriented activities were not outlined, contrasting with detailed plans for vaccines, diagnostics, global surveillance networks, and genomic sequencing capability (*Stern, 2021*). While detection and response measures tend to safeguard the privileged, prevention protects the most vulnerable from future crises.

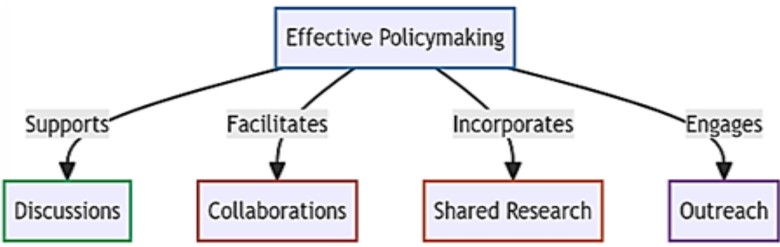

**Figure 4 Four foundational pillars crucial for effective policymaking.**

Under the International Health Regulations (IHR), the WHO defines countries' obligations to prevent the transnational spread of infectious diseases (*Baker & Fidler, 2006*; *Gostin, 2004*). However, the IHR primarily focuses on responding to emerging pathogens rather than addressing the drivers of disease emergence or providing guidance on upstream preventive measures (*Jones et al., 2008*). In response to the unprecedented threat posed by climate change and ecological destruction, there is a pressing need to transform our thinking and approaches to prevention, recognizing the inherent risks of the emergence of new pathogens (*Shanks, van Schalkwyk & Cunningham, 2022*).

The interconnection between spillover events and the emergence of new infectious diseases is undeniable, and it is anticipated that new pandemics, akin to COVID-19, will inevitably surface in the coming years (*El Amri et al., 2020*; *Tilak et al., 2021*). While the occurrence of such public health emergencies is an inherent aspect of human history, the probability of their manifestation can be mitigated (*Jit et al., 2021*). The severity of each pandemic is intrinsically tied to global unity and the equitable distribution of resources. Consequently, developed nations must allocate resources more equitably toward the development and well-being of undeveloped and developing parts of Asian, African, and South American countries, with a paramount focus on preventing spillover events that could have catastrophic global repercussions (*Shanks, van Schalkwyk & Cunningham, 2022*; *Sheikh et al., 2021*). A crucial shift in policy is imperative, emphasizing the prevention of future spillover diseases. The existing governance systems spanning human, animal, and environmental sectors pose a considerable challenge to achieving this paradigm shift. Revolutionary changes hinge on significant reforms within these organizations. Multi-stakeholder solutions are imperative to diminish and manage health risks associated with unsustainable commodity use, as well as those stemming from the capture, trade, farming, and consumption of wild animals and their products (*Pruvot et al., 2023*). To assist government prioritization of prevention, we highlight key actions that can be implemented.

### Safeguarding global health: proposing comprehensive strategies to prevent zoonotic spillovers and mitigate pandemic

In the pursuit of preventing zoonotic spillovers, we identify four foundational pillars crucial for effective policymaking: discussions, collaborations, shared research, and outreach (Fig. 4). To address existing policy gaps, particularly in undeveloped nations

requiring immediate attention, we propose targeted interventions. Our initial focus centers on educating the next generation and recognizing local communities as instrumental in reducing spillover risks. While education is a well-established avenue for disseminating information, we suggest the inclusion of a unique wildlife course in school curricula, starting from the 3rd standard to secondary school. These courses would emphasize zoonotic diseases, human-wildlife interactions, and global equitable development, fostering a comprehensive sustainable development vision among the younger generation. Importantly, it aims to educate students in developed European countries about the significance of impoverished communities in Africa, Asia, and South America to humanity's survival, fostering a reciprocal understanding.

Our second proposal involves the creation of a specialized task force on World wildlife prevention strategies explicitly designed for undeveloped and developing countries. This task force, funded by developed nations and overseen by experts such as epidemiologists and infectious disease specialists, would work on identifying spillover hotspots, upgrading infrastructure, and addressing local challenges. Recognizing that many policies fail at the ground level, particularly those centered on outlawing activities like bushmeat, hunting, or wild animal trade, we suggest a data-driven approach. The task force would collect data on individuals involved in hotspots, providing alternative employment opportunities and effectively closing policy loopholes.

Environmental sampling emerges as a critical tool in our preventive arsenal, alerting public health professionals to unusual disease outbreaks. This involves monitoring emerging infectious agents, understanding triggers for pathogen spillover, and utilizing field genomic data and mathematical projections to quantify disease burden and predict spillover occurrences. The subsequent step entails establishing a chain of integrated diagnostic laboratories and data collection centers. A central World wildlife disease laboratory monitoring system would integrate data from these labs, enabling a coordinated global response. District-level laboratories equipped with cutting-edge genomics technology and the ability to conduct serosurveys play a pivotal role in decentralizing surveillance efforts for faster detection of infectious agents.

While it may be impossible to entirely avoid emergencies in human history, collaborative efforts among governments, researchers, and health officials can minimize the risk. In the aftermath of COVID-19, we emphasize the enforcement of more stringent international laws regarding bush meat and wild animal product trade for traditional medicines. Lastly, inspired by the simplicity of the virus, we advocate for overcoming ethnic, religious, and nationalistic taboos, prioritizing humanity, and demonstrating our collective societal capabilities. The true failure lies not in our inability to prevent spillover events but in allowing them to escalate to pandemic proportions in the first place.

## CONCLUSION

In conclusion, the global challenges posed by zoonotic spillovers and viral mutations from LMICs demand urgent attention and concerted efforts to prevent future pandemics. The intricate interplay of environmental factors, human activities, and the increasing density of reservoir hosts in regions such as Asia, Africa, and South America elevates the

risk of zoonotic spillovers. The ongoing impact of zoonotic diseases on economic, social, political, and environmental systems underscores the necessity for comprehensive prevention strategies.

The review emphasizes the critical importance of surveillance as an early warning system to identify potential spillover events. However, the limited surveillance capacity in LMICs, as exemplified by historical outbreaks like the Nipah virus in India and Ebola in West Africa, underscores the need for global collaboration and investment in strengthening surveillance infrastructure. The WHO's international programs play a pivotal role in this regard, focusing on enhancing healthcare systems in LMICs.

The "vaccine gap" further exacerbates the threat by allowing unchecked viral replication and the emergence of novel mutants. Achieving global vaccination is not only crucial for ending the current COVID-19 pandemic but also for preventing future viral mutations. Equitable vaccine distribution, sharing surplus vaccines, and strengthening healthcare infrastructure in LMICs are imperative steps towards global unity in the fight against infectious diseases.

The analysis of international legal frameworks reveals both their significance in mitigating zoonotic diseases and the challenges in policy implementation, especially in regions where sustainable hunting is essential for livelihoods. Proposing comprehensive strategies, including education, specialized task forces, environmental sampling, and integrated diagnostic laboratories, forms the basis for effective policymaking.

In essence, the review advocates for a paradigm shift in prevention approaches, recognizing the importance of upstream measures and collaboration across borders. The proposed foundational pillars and targeted interventions provide a roadmap for policymakers, public health professionals, researchers, and global health stakeholders to work collectively in minimizing the risk of zoonotic spillovers escalating to pandemic proportions. The urgency of these collaborative efforts cannot be overstated, as the true failure lies not in the inability to prevent spillover events but in allowing them to escalate into global health crises.

### Funding
The authors received no funding for this work. Zulfiqar Ul Haq's postdoctoral fellowship was supported by the ICMR vide AMR/Fellowship/21/2022-ECD-II. The funders had no role in study design, data collection and analysis, decision to publish, or preparation of the manuscript.

### Grant Disclosures
The following grant information was disclosed by the authors:
ICMR vide AMR/Fellowship/21/2022-ECD-II.

### Competing Interests
Mudasir Ahmad Syed is an Academic Editor for PeerJ.

## Author Contributions

- Zulfqarul Haq conceived and designed the experiments, performed the experiments, analyzed the data, authored or reviewed drafts of the article, and approved the final draft.
- Junaid Nazir performed the experiments, analyzed the data, authored or reviewed drafts of the article, and approved the final draft.
- Tasaduq Manzoor performed the experiments, prepared figures and/or tables, and approved the final draft.
- Afnan Saleem performed the experiments, prepared figures and/or tables, and approved the final draft.
- H. Hamadani performed the experiments, prepared figures and/or tables, and approved the final draft.
- Azmat Alam Khan conceived and designed the experiments, analyzed the data, authored or reviewed drafts of the article, and approved the final draft.
- Sahar Saleem Bhat performed the experiments, authored or reviewed drafts of the article, and approved the final draft.
- Priyanka Jha performed the experiments, authored or reviewed drafts of the article, and approved the final draft.
- Syed Mudasir Ahmad conceived and designed the experiments, performed the experiments, analyzed the data, prepared figures and/or tables, and approved the final draft.

## Data Availability

This is a literature review.

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
