# Peer review of "Zoonotic spillover and viral mutations from low and middle-income countries: improving prevention strategies and bridging policy gaps"

_PeerJ, doi:10.7717/peerj.17394_

## Round 0.1 · original submission · Major Revisions

Although the topic of the review is extremely important, the reviewers found the manuscript overall disorganized, lacked of clear structure, conciseness, clarity, and correct references. In particular one reviewer felt that the isolated description of the Omicron lineage was not well developed. Another reviewer also found that the figures were not used in the corrected place, were incomplete and not clear. The reviewer have provided extensive comments to improve the manuscript.

Reviewer 1 ·

Basic reporting

This Reviewer was expecting a wide literature review about pathogens associated with zoonotic spillover vs different mutation patterns. As cited by the authors, “this review delves into the mechanisms of spillover and mutation, both in general and within the context of the developing world, highlighting the challenges, deficiencies in current global policies, and practical mitigation strategies tailored to these regions.” The authors initiated the work on the main zoonotic viruses and then went through specific genetic details about the Omicron variant, making it confusing to follow. In general, it would be interesting if the authors went through a well developed and supported argument, organized the text into logically and coherent paragraphs/subsections towards the goals/main points of the work. My additional impression is that the isolated description/mention about Omicron is vague and a bit confusing to follow and to link with the whole text. If this direction is the primary line and focus of the manuscript, I believe that the authors should rephrase the title to better reflect the topics described. The Reviewer also felt many references in various sentences (details can be found below), giving an impression of the formulation of their own point of view, which is somewhat paradoxical for a literature review.

Fig.1 - Would it be possible to improve the quality of the figure?

Fig.2 - What about the consequences of viral spillover from developed countries to undeveloped nations? See also Spanish (US), Asian (CH) and Hong Kong (HK/CH region) Flu, SARS-CoV/SARS-CoV-2 (COVID-19) (CH).

Experimental design

- The methods need to be described in sufficient detail in a way to be consistent with a comprehensive review of this topic.
- Another important point is about the range of years (time intervals) used for data collection (“in-depth searches”) in your study. Without this reference, readers cannot follow with precision what indeed was covered.
- About the range of years cited above, which pathogens did the authors find and what their main characteristics were obtained within the context of the study? What did the authors take from this in a spillover/pandemic context? What about the geographical movement of viral agents? Do they mostly come from a poor country to a rich one? Are they from rich nations? Maybe a detailed table showing these points could help improve the manuscript.

Validity of the findings

- Although the Reviewer understands the importance and meaning of the corrective measures (“foundational pillars”) proposed in topic 4.3, i.e., looking for the identification of unresolved questions and gaps and providing directions, their validity cannot be promptly evaluated. Due to their status involving complex spheres, it might take years to implement.

- Conclusions are well stated, but the authors should take into account the limitations of the placements and the comments above.

Additional comments

Introduction

Lines 62, 71, 82, 85, 92, 96 and 98: please, add references

101-102: Please consider removing “at the animal-human interface” since the zoonotic diseases term is already mentioning that.

Survey Methodology

- Explain why the authors cited specifically the Omicron variant. Seems out of context. If the manuscript is about CoVs, the title as well as other parts of the work must be rephrased, accordingly.

Line 112-14: Could you please provide a detailed table with such data?

1. Spillover and Surveillance

1.1. Spillover Mechanisms and Risks (additional observation - very long paragraph)

Line 121: Reference

1.2. 1.1 (typo here? - 1.1). Significance of Surveillance (additional observation - very long paragraph)

Line 142: Same as lines 101-02.

2.2. Entry Pathways and Future Concerns

Lines 223, 226, 229, 233 and 249: Please, add References

4. Spillover of Zoonotic Diseases: Current Policies and Deficiencies

Line 329: Reference

4.3 Corrective measures: A call from undeveloped countries

This Reviewer feels that a schematic figure/fluxogram showing the 4 foundational pillars proposed could benefit the readers as well as a broader audience.

Line 410: the virus "Be Simple" by simplicity?

Reviewer 2 ·

Basic reporting

In this review, the authors described the need for understanding historical factors that contribute to spillover events and called the attention for better preventive measures mainly in developing countries. The topic of this review is extremely important, as the world faces the threat of new pandemics. The authors gave important suggestions to improve preparedness for a new pandemic, particularly by increasing education of the new generations and by appealing to a share of excess of vaccines with the developing countries. This review is indeed of cross-disciplinary interest and within the scope of the journal, and is accessible to a wide audience (i.e., not just a scientific audience).

However, this review is disorganized, it’s not clear and its very repetitive. There are also some statements that lack reference support. In addition, the figures are only used once, not used in the corrected place and are incomplete and confuse.


Major changes related to the Basic reporting:

1 - The introduction needs to be rewritten, as it does not introduce the topic well, and it’s hard to follow up. There is no order for the background provided. I give some examples below.
Ex.: Line 49: ‘The risk of these zoonotic infections is higher than ever, with three out of four new diseases being zoonotic [2]’. Then you return to the same topic introduced beforehand in lines 56-59.
Then in line 65 the authors refer about vaccine vacuum without previously introducing and defining it previously. Then the authors return to it on line 89. It’s important to be consistent – vaccine vacuum or gap.
The authors started paragraph on line 69 correctly referring about causes for the increase contact between humans and animals (potential reservoirs of zoonotic diseases). However, they return to talk about the same idea as between lines 53-54. Again, the idea is there but it’s repetitive and not clear.
2 – Lines 76-77: And what about the south American continent? South America is the home of the largest forest in the world, with an incredible biodiversity, and it’s being identified as the next possible origin for a pandemic. For example, you can take a look at this https://www.science.org/content/article/scientists-scour-amazon-pathogens-could-spark-next-pandemic and https://www.conservation.org/blog/study-could-the-amazon-become-ground-zero-for-the-worlds-next-pandemic
3 – Lines 80-88: I would advise to rewrite this paragraph without mentioning ‘China’ directly. It’s not good practice. Besides, there are other markets for the consumption of bushmeat and wildlife trade. The authors make different affirmations in this paragraph with citing only one reference at the end.
4 - In introduction, the paragraph between lines 89-107 needs relevant references, for example in lines 94-96 and 96-98.


Regarding the general structure of the manuscript, it should be improved for clarity.
For example, the manuscript starts very well in the abstract stating the need to better understand zoonotic disease spillovers in general, but then in section 2 the authors deeply address the Omicron variant, without a previous introduction to the topic and why this serves as example. In the section about vaccines, the only example is about the SARS-CoV-2 variant. The manuscript does not follow a sequence of ideas and the text does not flow. Definitively this manuscript needs more references. It’s frequent that the authors do more than 2 statements and only use 1 bibliographic source.

Experimental design

Some points of the study design have been addressed previously.
The manuscript content is within the aims and scope of the journal. However:
1 - The authors claim that they did a rigorous investigation, but don’t explore it further (don’t provide details in how that has been done). The methods were not described in detail. How many papers were accessed in total, and from how many countries/continents? Which search engines were used? How can we know that this is not an unbiased coverage of the subject? The paragraph between lines 109-114 is not informative. The authors need to describe all the key words and search engines used, how many papers were accessed, covering each countries/continents, and years.
In addition, several statements in the text don’t have a reference. A scientific review needs references. If not it’s just an opinion. I have pointed this out below as minor changes.

Validity of the findings

In general, it’s not clear what the paper is about. Is it about the concern with spillover of zoonotic diseases or just the emergence of new SARS-CoV-2 variants? The authors don’t follow a sequence for the sections. It’s disorganized. It seems the authors use the COVID-19 pandemic just as an example, for new pandemics, but that it’s not clear.
1 – In lines 142-148 the authors propose a two-step approach to address current deficiencies in policy, but don’t develop it further here. It’s only in section 4.3 that this approach is identified.
In addition, this whole section about surveillance should be reduced and clarified. The authors repeat information that surveillance is needed. But why it’s not working? Are countries not doing surveillance? Is there no surveillance happening in the developing and undeveloped countries? Could you give some examples? Regarding the needs for surveillance and research on Neglected Tropical diseases, not only in new emerging pathogens, in developing and undeveloped countries, the authors could also explore further the international cooperation (e.g., how international cooperation should work). More than just funding from developed countries. There is a lot of international cooperation happening right now. I know that in face of so many needs, it’s never enough. If this international collaboration is not working, the authors should identify what is been done (with references) and then, what is missing.

Section ‘Global Surveillance Initiatives’ – could be rewritten as a summary about the strategies of WHO, FAO and WOAH. What are the strategies and why are they not successful?
Section 4.3 – Interesting ideas but: lots of it is already being done. Too much detail in areas that don’t need. No bibliographic references supporting the affirmations. Although I recognise the importance of this topic, the urgency of new measures and the power of education, this manuscript is more an opinion paper rather than a scientific review. The authors repeat a lot the same information.
Also, could the authors list any examples of research in developing/undeveloped countries. For examples, you could cite the works from Professor Katie Hampson in Tanzania, about rabies dynamics, and vaccination strategy and surveillance design. This is just an example, but there is a lot of good research happening in the world that the authors don’t address.

The manuscript does not provide enough support to answer to the abstract or the conclusions.

Additional comments

Minor changes:

Check all manuscript:
Inconsistence: Bush meat should be bushmeat.
Check manuscript for no space between word and reference.

Lines 26-27: The authors state on the abstract and later, on the manuscript, that zoonotic disease spillover events are on the rise particularly in less developed Asian and African countries. The authors should have references that support this affirmation. Of course, not in the abstract, but on the manuscript.

Why just Asia and Africa? What about the south American continent with the biggest forest in the world – Amazon?

Introduction:

Line 47: define ‘SARS’ as it is the first time you are referring it.
Lines 49-50: the sentence does not make sense; rephrase it. Maybe: the risk of zoonotic infections …
Line 53: biofluids
Lines 55, 79 and 107 and respective figures: be consistent Fig 1 or Fig.1 or fig 1
Line 58: define HIV
Line 65: briefly explain the concept of ‘vaccine vacuum’ between round brackets, as it is the first time you refer to it.
Lines 65-66: the statement here needs a reference.
Line 73: define WHO
Line 89: be consistent throughout all manuscript: vaccine gap or vacuum?
Lines 89-107: This needs a reference.
Line 103: increase letter size ‘may’.
Line 109: Have you not included spillover, and mutation in your search?
Line 113: define FAO. Currently is not OIE anymore but WOAH – World Organisation for Animal Health. You have it in line 298
Throughout the manuscript: define one abbreviation the first time you referred it in the text. Be consistent. Another example are lines 299-304/305.
Line 117: remove ‘and infections’
Line 140: Remove full stop before the reference.
Line 142: remove 1.1.
Lines 142-148: Change this sentence. This is exactly what you have in the abstract.
Line 160: Remove full stop (.) before reference [19] – check this consistently throughout the manuscript.
Line 160: Is reference 19 adequate for all the statements between lines 154-160?
Lines 163-164: No need to define WHO again as you have refereed it before.
Line 173: Again, not OIE anymore.
Lines 173-186: The authors started the paragraph talking about global surveillance initiatives, to then on line 178 talk about how animals can be good surveillance indicators. They should link these two ideas and provide more references for the statements. Information between lines 178-186 can be summarized.
Lines 187-189: As this is a scientific review, some statements as this one, would benefit of more original references. Often the authors use other reviews as references and don’t go to the original source. Between lines 187-197 the authors give examples of when animals are used as sentinels only citing 2 references.
Line 191: ‘Anomalocarids brasiliana can predict’ – change for can be used to predict.
Line 193: Bacillus anthracis should be in italic
Line 194: Yersinia pestis should be in italic – please confirm all manuscript other cases that should be in italic.
Line 203: space between ‘contact’ and reference. Check the manuscript because this happens another time.
Line 213: NTD and FCS need full stop?
Lines 256-264: The authors should add a reference that supports these statements.
Line 273: Define CEPI and Gavi
Line 280: Full stop between [38] and Efforts.
Lines 284-285: remove ‘and requested’.
Lines 294-305: This paragraph does not say much about international legal frameworks. There is just 1 supporting reference here that it’s from 1 workshop report.
Line 302: it is believed that increased interaction??? Interaction between what?
Line 303: Again – why just Asia and Africa? Where are the stats and the references to support this information.
Line 314: Add full stop between pandemic and Approximately
Lines 316-319: Add reference
Lines 332-335: After a lot of text, the solution provided is: ‘the developed European nations must put more resources equitably toward the development and welfare of poor Asian and African countries’. The authors forgot other countries in other continents.
Lines 328-413: There are no references for all these statements.
Lines 356: focusingon – should be focusing on
Lines 382-384: Is environmental sampling not happening now? There is lots of sampling going on.
Lines 415-416: Change ‘correlation’ (as you have not assessed the correlation) for association or another synonym.

References list: There are some inconsistences. Ex., ref 4, 19, 32, 35, 36, 39, 40,
Also, most references don’t have a DOI.



Figures:
The figures should be redesigned.

Figure 1: Legend should be:
Major pathways to zoonotic spillover
This figure is incomplete and it’s not clear. For example, check Plowright et al., 2017 (Nature)


Figure 2 is not clear, and it is not needed because does not add much and is incomplete. Figure 3 has poor quality. However, it has value because it tries to explain the two-step approach, hence should be redesigned, and clarified.

---

## Round 0.2 · Minor Revisions

Although reviewers acknowledged that the review has improved considerably, there are still some points that need to be improved. Reviewers in particular have requested figures to be of higher quality, to re-numerate sections more consistently, and many other details that require further attention. Please read the comments of the reviewers for more details.

Reviewer 1 ·

Basic reporting

No comment

Experimental design

No comment

Validity of the findings

No comment

Additional comments

Topic: Survey Methodology
Line 144: authors until

The figures also need to be improved (HQ)

Few corrections

Fig. 1: contaminated environment
Fig. 2: field 1 - global equitable
Fig. 2: field 2 - strengthing surveillance; possible spillover; prioritising strategies...

Reviewer 2 ·

Basic reporting

This review has improved considerably. The text its clear although a bit dense. Supporting references have been added when needed. However, there are still some minor points to check. The line numbers below refer to the clean version of the manuscript.

General: the numeration of your sections is confusing because you put for example 2.2, then 1, 2, 3, 4, or even more confusing when you pass from a 4 to a 3 again. In this case, the subsections should be 2.2.1, 2.2.2, 2.2.3 (…). You need to clearly numerate your subsections. The subsections should be consistent – such as always using A., B. or a) and b). Then you have 2.1 as ‘Importance of Global Vaccination’; here is section 2.?

Be consistent in your abbreviations: For example, use I.H.R. or IHR as you use WHO and FAO. Correct all manuscript.

Experimental design

The article content is within the Aims and Scope of the journal. The authors have improved the description of their methods.

Validity of the findings

No comment.

Additional comments

Line 29: ‘bush meat’ change to ‘bushmeat’ – according to Oxford dictionary, bushmeat is correct. Please change accordingly here and throughout the manuscript to be consistent: lines 76, 119, 532, 546.
Lines 52-54: remove this repetition of the previous sentence.
Line 74: You defined here World Health organization (WHO) for the first time, and you don’t need to repeat the definition again in the manuscript. Example, delete ‘World Health organization’ and keep the abbreviate name (WHO) in lines: 261, 284, 301, 309, 316, 323, 338, 430, 490.
Lines 75-76: wild meat is the same as bushmeat. Remove wild meat.
‘The term 'wild meat' (or 'bushmeat' in Africa) refers to terrestrial wild animals used for food in all parts of the world (IUCN World Conservation Congress 2000).’
Line 78: the consumption of wild meat is also popular in indigenous communities in Amazonas (south America).
Line 112: here you can delete ‘in Asian and African countries’ and just leave it as ‘bushmeat hunting’ or add ‘south American’ as later in line 119 you refer bushmeat in poor Asian, African, and South American countries.
Lines 129-133: the sentence does not read well. Change for: ‘Supplementary data were gathered from country reports, newsletters, commentaries, and policy briefs from Food and Agriculture Organization (FAO) and World Organisation for Animal Health (WOAH), because it was determined that some of the relevant sources were (…)
Also, here is the first time you define FAO and WOAH; hence you don’t need to use their full name in the next text. Delete ‘Food and Agriculture Organization’ or ‘World Organisation for Animal Health’ from lines 338, 339, 460.
Line 135: until and not untill
Line 146: ‘immune defences’ and not defenses.
Line 154: Remove the abbreviation (ACE2) as you are not going to use it further.
Line 183: ‘Henipavirus genus’ – ‘Henipavirus’ in italic
Line 202: H5N1 change for H5N1
Line 246: remove (CNS) as you are not going to use it further.
Line 248: Need a full stop after 20 years.
Line 262: you need to define low and middle-income countries the first time you use it – in line 83. Afterwards, only use the abbreviated format (LMICs). Change throughout the manuscript. Example: lines 262 and 438.
Line 292: remove (NICs) as you only use it one. Could you please check the manuscript and remove all abbreviations just used once?
Line 355: change for H1N1 and H5N1.
Lines 376-377: I would remove from the title ‘Asia and Africa’ and leave it as general ‘low-income countries’, as this can also be applied to some Latin American countries.
Line 404: Full stop after ‘across borders’.
Lines 421-427: Remove this paragraph as it is repeating what you said in lines 397-402.
Line 461: remove ‘(W.O.A.H., formerly O.I.E)’ as you don’t need it.
Line 470: Remove ‘Intergovernmental Science-Policy Platform on Biodiversity and Ecosystem Services’ and use its abbreviation.
Lines 502-503: ‘Consequently, developed Asian and European nations’ and the USA are not developed? Or Australia? Please remove the reference to ‘Asian and European’ and keep ‘developed nations’.
Line 516: Fig 4
Line 558: instead of ‘correlation’ use ‘relationship’. The sentence is describing a relationship or connection between two variables: spillover occurrences and the emergence of novel contagious ailments. Using "correlation" suggests that there is a statistical association between these two factors, which you have not assessed via a statistical test.

---

## Round 0.3 · accepted · Accept

After have assessed the revision, I confirm that the authors have addressed all of the reviewers' comments in the current version which I find very much improved and ready for publication.